# Poka Yoke Meets Deep Learning: A Proof of Concept for an Assembly Line Application

Matteo Martinelli *[ID], Marco Lippi [ID] and Rita Gamberini [ID]

Department of Sciences and Methods for Engineering, University of Modena and Reggio Emilia, Via Amendola 2, 42122 Reggio nell'Emilia, Italy; marco.lippi@unimore.it (M.L.); rita.gamberini@unimore.it (R.G.)
*   Correspondence: martinellim@unimore.it

**Abstract:** In this paper, we present the re-engineering process of an assembly line that features speed reducers and multipliers for agricultural applications. The *"as-is"* line was highly inefficient due to several issues, including the age of the machines, a non-optimal arrangement of the shop floor, and the absence of process standards. The assembly line issues were analysed with Lean Manufacturing tools, identifying irregularities and operations that require effort (Mura), overload (Muri), and waste (Muda). The definition of the *"to-be"* line included actions to update the department layout, modify the assembly process, and design the line feeding system in compliance with the concepts of Golden Zone (i.e., the horizontal space more ergonomically and easily accessible by the operator) and Strike Zone (i.e., the vertical workspace setup in accordance to ergonomics specifications). The re-engineering process identified a critical problem in the incorrect assembly of the oil seals, mainly caused by the difficulty in visually identifying the correct side of the component, due to different reasons. Convolutional neural networks were used to address this issue. The proposed solution resulted to be a Poka Yoke. The whole re-engineering process induced a productivity increase that is estimated from 46% to 80%. The study demonstrates how Lean Manufacturing tools together with deep learning technologies can be effective in the development of smart manufacturing lines.

**Keywords:** Lean 4.0; Poka Yoke; smart factory; CPS; First Time Quality; convolutional neural networks; assembly line

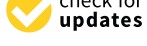



## 1. Introduction

The Poka Yoke concept was first formalised by Shigeo Shingo [1] within the context of the Toyota Production System [2]. The Poka Yoke idea lies in geometrical or methodological constraints aimed to prevent unwanted operations or events. Some examples are given by the component positioning of a mechanical assembly constrained by geometry (geometrical Poka Yoke) or tools that constrain certain actions in order to achieve or not a given effect (methodological Poka Yoke). An example of the former is the USB-C connector, whereas an example of the latter is the clutch mechanism in the last generation manual transmission vehicles with the aim of starting the engine and preventing accidental movement of the car during its ignition.

In manufacturing contexts, a Poka Yoke is a solution whose goal is to prevent human errors through a process constraint design (generally low cost). All market solutions are oriented to obtain zero defects: the final target is to avoid low quality products reaching the client due to incorrect processes. Industry 4.0 brings new technologically advanced possibilities also in this area. In particular, the contamination of Industry 4.0 and Lean Manufacturing has led to the so-called "Lean 4.0": under this lens, Lean Manufacturing becomes a prerequisite for the application of the concepts of Industry 4.0, and Industry 4.0 in turn promotes Lean Manufacturing [3].

A strict relation between Lean Manufacturing approaches and Industry 4.0 technologies has been proved also by [4] via a methodological survey proposed to sector experts.

The result highlights how many of the Lean Manufacturing tools (JIT, TPM, Automation and VSM) can benefit IoT applications, improving information flow, decision making, productivity, and responsiveness. The positive relation between Lean Manufacturing and IoT makes the latter the basic building block of Industry 4.0, connecting the *Gemba*-physical world to the digital world and enabling advanced applications.

In this paper, we present the re-design of an assembly line with a process that combines methods that follow typical lean production principles and novel techniques based on artificial intelligence and computer vision. During the re-design of the assembly line study, a practical problem linked to a human assembly error emerged, namely the correct positioning of an oil seal within a speed increaser. The problem affected the First Time Quality index of the assembly line, lowering its overall performance and generating frequent returns in customer orders. This problem needed a careful analysis and a customised solution exploiting advanced deep learning technologies applied to a computer vision problem. The overall solution resulted in a Poka Yoke that impacts positively on the First Time Quality production index.

The paper is organised as follows. Section 1.1 describes related works in the context of artificial intelligence applied to Lean Manufacturing. Then, Section 2.1 illustrates all the phases of the redefinition of the assembly line, whereas Section 2.2 focuses on the problem of the automatic control of the oil seal placement and describes the proposed solution based on computer vision tools. Section 3 depicts project results, and Section 4 concludes the paper.

### 1.1. Related Works

The enabling technologies offered by the fourth industrial revolution—IoT, Big Data, and Machine Learning—carry within them the potential to redefine the Poka Yoke concept, moving from mistake proofing systems to assistance systems for line operators. Intelligent Poka Yoke tools have the potential to lower around 33% of human controls performed along assembly lines, which are often repetitive and redundant [5]. Among the most promising technological tools are augmented reality (AR) systems and systems based on computer vision and artificial intelligence. As an example, AR is gaining attention not only for applications in the assembly and maintenance areas but also for Quality Control tasks [6].

However, as discussed later, even applications based on simpler and different types of sensors (such as microphones, vibration sensors, force sensors) can lead to excellent results if supported by the possibilities offered by Industry 4.0. A great deal of practical applications of these technologies have already been studied and have already begun to spread in the most advanced industries. In the following, we describe three main groups of related works.

#### 1.1.1. Equipment Control

In the application domain of end milling, a solution to perform condition monitoring on cutting tools is presented, by exploiting a microphone whose data are processed by a convolutional neural network (CNN) [7] . The project implements a detector for the state of the tool in use to alert the operator before an imminent breakage. Other applications of modern technologies to Computer Numerical Control (CNC) machine monitoring have been proposed by [8], who apply deep learning techniques to data collected by an accelerometer positioned on the tool head in use. CNCs are the object of study also of [9], which implements an early detection machine failure based on vibration signals. The system implements two components: a signal analysis system, based on a fuzzy neural network, and a decision maker, whose goal is to classify the machine state as normal or in alarm or to shut it down More generally, [10] analyses the methodology that should be carried out in order to retrofit or connect existent CNC machines in order to make them smart and fool-proof, with a special attention to human factors. The authors of [11] consider a generic maintenance process, which can be potentially applied to any workstation composed of

machines or mechanical or electrical equipment. The goal of the project is to standardise the data acquisition phase nearly after the breakdown occurrence in order to improve the data quality and consistency, avoiding missing or incorrect parameters during the fault description. As a consequence, average maintenance scheduling lead times have been drastically reduced, resulting in better data organisation and therefore better performance tracking. In [12], the authors implement a system that is able to automatically generate customised CNC programs compliant with specific requirements. The need for such a system lies in the constantly increasing demand for customised products, which makes it challenging to have a manual CNC setup for each of them. The outcome of the study is a system based on decision trees.

### 1.1.2. Quality Control

Another application related to the analysis of information collected through a microphone is described by [13]. The idea was applied in the context of the quality control of a differential, where the sound emitted by the machine during the test is analysed to detect whether the component is functional. In [14], on the other hand, recurrent neural networks (RNNs) and CNNs are applied to frequency signals obtained from a vibration sensor. Data are pre-processed with the Short-Term Fourier Transform (STFT) in order to correctly classify the health status of a ball bearing. In [15], the authors also focus on analysing the vibrations generated by a broken ball bearing in frequency, but applying the S transform instead of the STFT in pre-processing. Another interesting proposal is the one developed by [16] which implements an X-ray data collection method, with the images then manipulated through the use of computerised tomography and CNNs. The ultimate goal of their project is to understand whether the internal components of a finished and hermetically sealed product have been assembled correctly. In [17], the usefulness brought by a commercial 3D printer in the quality control area is explored, following the Poka Yoke spirit. It underlines the capability of 3D printers of producing highly customised, cheap tools that are useful for quality checks if combined with the Poka Yoke methodology. The impact on associated process performances are clearly pointed out, as well as the impact on customers.

### 1.1.3. Process Control

CNNs and RNNs are also studied by [18], with the goal of minimising the pre-processing of time series of data in complex system. The system is then validated with an experiment carried out on a service lift whose various types of anomalies are to be predicted. Machine learning tools are also applied for the purpose of monitoring the manufacturing process of robotic cells, which are automated in order to classify their output and possibly to set the parameters on-the-fly. A contribution with a system applied to laser additive manufacturing cells is made in [19]: the data are collected during the process from a low-cost coaxial camera and are processed by a CNN in order to detect the porosity of the artefact. Another example of process monitoring is that offered by [20]: they collect the process information of a hybrid laser–MAG (Metal Active Gas) welding system with two cameras at a high frame per second rate and extract the information through a convolution with the Gabor filter. A monitoring system for a robotic arc welding process is also proposed by [21]. Their solution is based on an optical system designed specifically for the application , whose images are processed by a CNN in order to classify the welding quality operated by the robot. Among the systems operating in real time, the insertion of an adaptive CNC machine parameter solution trained with neural networks to obtain the surface roughness within a desired value in an in-process manner is done by [22] Similarly, a controller applied to an injection moulded recycled plastic moulding system is implemented in [23] . Data collection is performed by an accelerometer, while data are processed using a classifier based on logistic regression.

The approach described in [24] implements an integrated environment based on AR and force sensors. AR was tested through the use of two types of equipment: the first

includes a fully immersive viewer, which reports the surrounding reality through the use of a camera, with information projected over the images reproduced by the viewer; the second, on the other hand, simply consists of a camera designed to monitor the work bench and a screen on which information is presented. The machine's understanding of the surrounding world is operated by tags positioned on the work bench. In [25], the authors developed a glove capable of monitoring the force and vibrations resulting from connecting two electrical terminals to understand whether the task was successful or not. AR technology has been also applied by [26]. In this article, AR was selected as a solution in order to facilitate new workers in carrying out activities requiring high skills, relying less on specialised training and, at the same time, reducing production scraps generated by human errors. As a result, 96% of users reported their interest in the tool.

Returning to the simpler and less invasive Poka Yoke, we cite the proposal of [27], consisting of an analogical camera, an analogical-to-digital converter, and a computer based on an ARM processor. Their solution saves the image in binary format, to be then processed and classified by a Hopfield neural network (a declination of the RNN). The example studied by [28] also exploits computer vision. The authors' goal was to replace human operators during the visual inspection of product quality. The implemented system is based on a low-cost device (a Raspberry Pi) connected to a camera, which analyses the portion of the image in which the characteristic to be evaluated should be captured. The system was applied and tested in a processing cell with the aim of estimating in the aftermath the correct positioning of a component before being processed. Another simple and low-cost solution is the one designed in [29], the goal of which is to operate a fault detection and identification (FDI) in an O-ring positioning machine. The designed system consists of a grey-scale sensor (composed of a photo-resistor and an LED) and two switches designed to send the presence signal of the O-Rings to the processing machine. The data acquired through the sensors are then processed by an artificial network trained with a non-supervised method, and the results are then compared with a traditional rule-based method. A recent work provided by [30]proposes a computer vision-based solution that monitors and evaluates operator assembly activities in order to identify production errors. The computer vision technique was chosen in order to prevent non-negligible investments in sensors and additional industrial equipment. Deep learning technologies were used, and a Manufacturing Description Language was developed. The output solution is able to provide recommendations to assembly operators based on a video stream and on the ongoing activities with respect to the planned chain of actions established by the management. Moreover, the system also has the capability to understand if the process has been completed correctly or if some mistakes have been made in the process. In [31], the authors focus their proposal more on the fully integrated autonomous factory model applied to the polymer industry, implementing a system that is able to autonomously adapt a high-level production plan through a distributed system. This is made possible thanks to distributed intelligent sensors and actuators represented in the virtual environment by agents. The resulting cyber-physical system enables the possibility to continuously monitor production deviation, adjusting needed parameters in real time.

*1.2. Bibliography Summary and Project Positioning*

This project proposes a Poke Yoke that leverages the enabling technologies offered by Industry 4.0: the solution consists of a simple camera used to recognise what is happening by framing the subject of interest and by monitoring it from different positions. Specifically, the final application sees the operator directly involved in the cycle: in the event that what is resumed does not coincide with the desired process, the process performed by the operator is made unavailable, reactivating its availability only after fixing the process. In the following, the innovation of the solution proposed in comparison with literature and practice results is deeply analysed. The main contribution with respect to the analysed literature consists in the fact that the solution is *proactive* and *physical* with an affordable cost, resulting in a best in class solution for the described problem in the Lean Manufacturing field.

Using the definitions of [1,32], we can now categorise the solutions described above according to the following characteristics:

- Sensors: The main sensors used to implement the solution; for the solution to be robust, they must be few, cheap, and resistant to a hostile environment;
- Data Processing technique (DP): The processing technique for the data collected by the sensors; for example, using machine learning approaches.
- Proactive or Reactive (P/R): As proposed by [1], a proactive Poka Yoke avoids the onset of the defect, whereas a reactive Poka Yoke performs detection without prevention.
- Physical, Functional or Symbolic (P/F/S): The definition proposed by [32] and cited by [33] whereby a physical Poka Yoke blocks the flow of mass, energy, or information without being dependent on an operator's interpretation; a functional Poka Yoke can be activated or deactivated following an event without depending on the interpretation of the operator, while a symbolic Poka Yoke can require an interpretation in accordance with the situation. This definition is again used to classify both technologies explicitly classified as a Poka Yoke by the authors and technologies of various nature monitoring quality parameters and ensuring the management or elimination of defects.
- Human-in-the-Loop (HIL): Whether human operators are involved in the operations.
- Cost: The expected cost depending on the technologies and solutions used.

All the solutions presented above attempt to resolve quality problems through tthe imely interception and reporting of the errors or by eliminating the occurrence of the errors at the source; however, only a portion of them have been designed with the specific intention of generating a Poka Yoke. Rather, technological solutions of various nature monitoring quality parameters that can generate errors or rejects have been presented. Following such a perspective, the mentioned solutions are listed and compared in Table 1. The solution proposed in this paper is indicated in the last line of the table, characterised as defined above. It emerges as a Proactive and Physical Poka Yoke.

**Table 1.** Existing Poka Yoke approaches comparison, exploiting machine learning and computer vision in the industrial domain. DP stands for Data Processing technique, P/R for Proactive or Reactive, P/F/S for Physical, Functional and Symbolic, and HIL stands for Human-In-the-Loop.

| Reference | Sensors | DP | P/R | P/F/S | HIL | Cost |
|---|---|---|---|---|---|---|
| [7] | Microphone | CNN | R | S | No | Low |
| [13] | Microphone | CNN | R | S | No | Low |
| [8] | Accelerometer | CNN | R | S | No | Med |
| [14] | Vibration | RNN+CNN | R | S | N/A | Low |
| [15] | Vibration | RNN+CNN | R | S | N/A | Low |
| [18] | 20 different sensors | RNN+CNN | R | S | No | High |
| [19] | 395fps camera | CNN | R | S | No | Med |
| [20] | 2000fps camera | Gabor Filters | R | S | No | Med |
| [21] | Modified camera + ad hoc mirrors | CNN | R | S | No | High |
| [22] | Vibration | CNN | P | F | No | Low |
| [23] | Accelerometer | RegLog | P | F | No | Low |
| [24] | AR + force sensors + cameras | N/A | R | S | Yes | High |
| [25] | Wearable vibration + force sensors | ANN | R | S | Yes | High |
| [27] | Camera | Hopfield RNN | R | S | No | Med |
| [28] | RaspberryPi and camera | Similarity Index | R | S | No | Low |
| [29] | Grey scale and limit switches | ANN + Rules | R | S | No | Low |
| [16] | X-Ray Camera | CNN | R | S | No | Low |
| [30] | Camera | DNN and MDL | R | S | Yes | Med |
| [9] | Vibration | FNN | R | F | No | Med |
| [10] | Raspberry Pi and general hardware | Heterogeneous | R | N/A | Yes | Med |
| [11] | Existing + retrofitted | DB | R | S | Yes | Med |
| [12] | N/A | Decision tree | P | P | No | Med |
| [17] | N/A | CAD/CAM | R | P | Yes | Low |
| [26] | AR | N/A | P | S | Yes | High |
| [31] | IIoT | Agents | P | F | No | High |
| Our approach | Camera | CNN | P | P | Yes | Med |

## 2. Materials and Methods

### 2.1. Assembly Line Redefinition

The considered scenario consists of an assembly shop floor, where the assembled product is a speed increaser, a mechanical instrument that finds its main application in the agricultural field, generally on towed equipment (for example, hydraulic gear pumps). Its function is to adapt the motion of the tractor power take-off output to the equipment input. The product consists in a crankcase with a cover; two input bearings and one oil seal; two output bearings and one oil seal; two centring pins; three oil plugs for oil load, unload, and inspection; one input shaft equipped with one O-ring and one plug; one output shaft equipped with one key, one Seeger ring, and one ring gear; eight closing screws; and one gasket. All the components are depicted in Figure 1a.

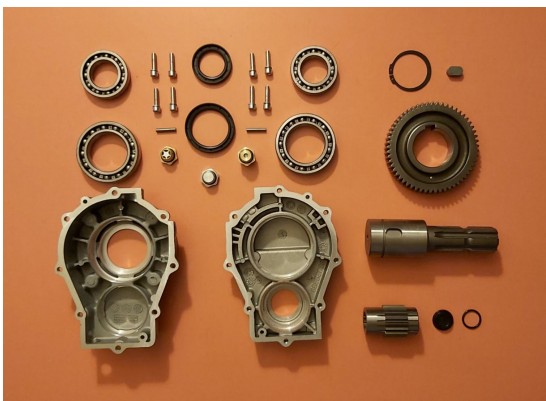

(**a**)

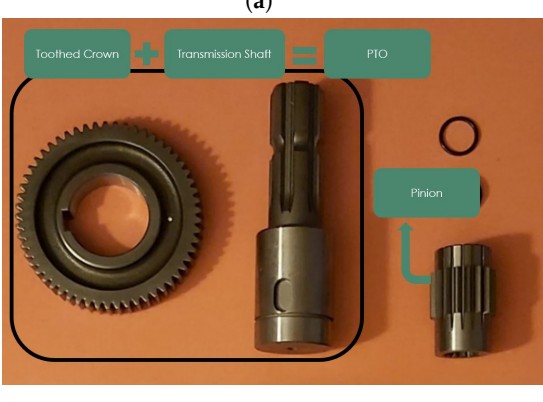

(**b**)

**Figure 1.** An overview of the product components (**a**). In (**b**), From the right: toothed crown, transmission shaft, and pinion. The toothed crown and the transmission shaft are coupled by interference, composing the power-take-off (PTO).

The internal gears are made up of toothed crowns joined with interference to the relative transmission shaft or directly obtained on the shafts themselves. The shafts are differentiated into "pinions" and "power take-offs": the pinions are the shortest shafts, whose reduction teeth are generally obtained directly on them; power take-offs, on the other hand, are the "longer" shafts, which generally join by interference to the relative crown. All the pieces are shown in Figure 1b.

The power take-offs and pinions run on ball bearings in an oil bath, and the tightness of the system is guaranteed by double or single lip axial oil seals, depending on the model. The product unit is enclosed within two shells obtained by fusion (main materials are aluminium or cast iron): the higher shell is called the cover, while the lower one is called the crankcase.

### 2.1.1. The "As-Is" Analysis

The analysis of the existing assembly line was based on the following steps:

1.  Choice of model product—that is, the product that involves the highest level of resources or that brings the highest contribution to annual revenue (crucial choice because of the high number of finished products – almost 200 codes in the catalogue; to produce a significant improvement, it is necessary to choose as a reference the model that impacts the most on the business).
2.  Layout analysis, which maps the actual disposal of the existing infrastructure in the "as-is" state.
3.  Material flow analysis, which maps the material flow between the areas and the other plant infrastructure in the "as-is" state.
4.  A spaghetti chart, which maps the assembly operators moving in the area in the "as-is" state.
5.  Mura analysis, which identifies and classifies all the wastes that emerged in the previous analysis.
6.  Limits of actions, to analyse the constraints given by management in the project implementation.

Choice of the Model Product

In order to select the most suitable product for the analysis, an extraction from the factory database was performed. This analysis showed that more than 190 different codes were ordered in the past three years. In particular, 20,216 units were sold in 2018. Sorting them by volume according to a Pareto diagram, we obtained the chart displayed in Figure 2. The diagram shows that the first 30 codes alone led to the production of 80.46% of the total volume. The code that received most of the orders is the first one, with an impact of 11.83% over the total annual volume. Such an impact is confirmed also by the revenue prospect: the graph displayed in Figure 3 reports the annual revenue per code maintaining the Pareto volume diagram order. Therefore, the product associated to that code was selected for our analysis.

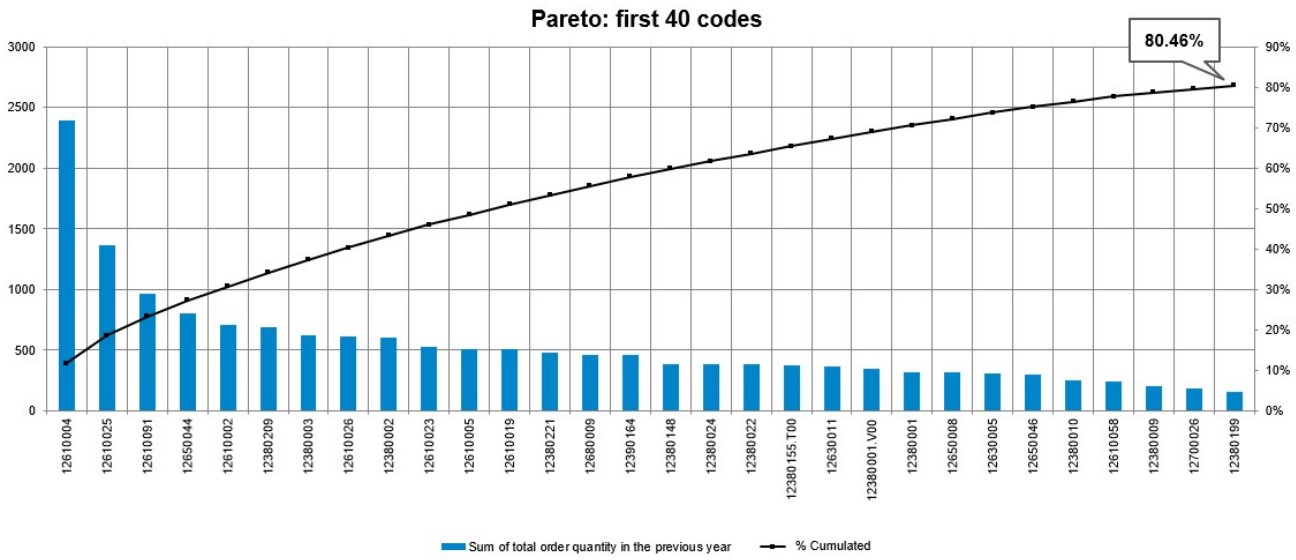

**Figure 2.** Pareto graph representing the 40 codes corresponding to the most ordered products in the current year.

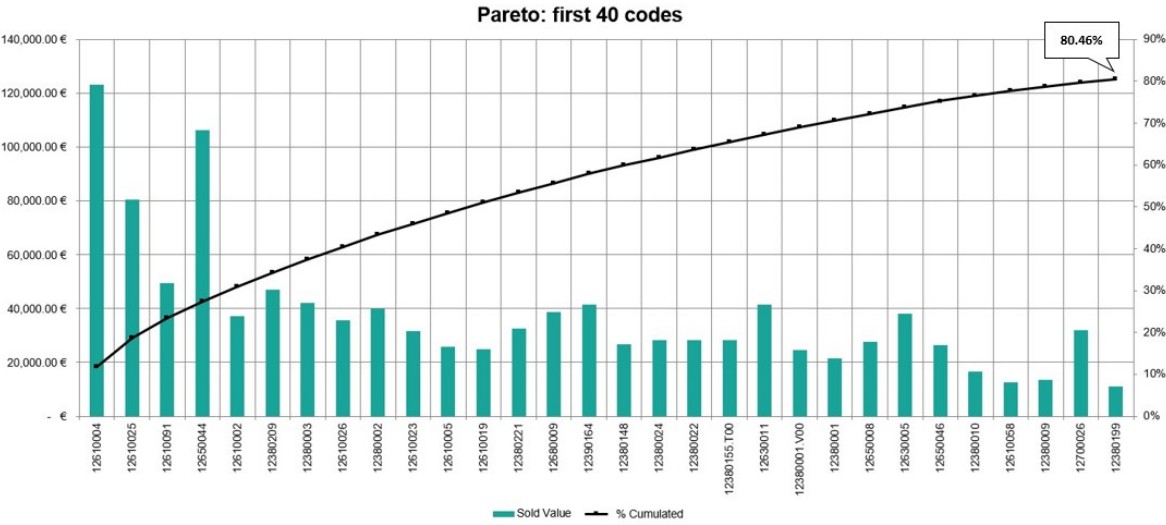

**Figure 3.** Pareto graph of orders made, expressed in Euros.

Layout Analysis

The existing production line is represented in Figure 4.
The area was characterised by three assembly islands:

1. Island F301, where the crankcase and cover sub-groups are pre-assembled.
2. Island F306, where the pinion sub-groups are pre-assembled.
3. Island FIS9, where the multiplier assemblies are completed, and which is further divided into two sub-areas: area A, where assemblies are completed only for small-to-medium multipliers; area B, where the assembly of large multipliers and, if necessary, also of medium-to-small multipliers is performed.

The feeding areas were scattered, and the feeding logic was not defined. The feeding structures were characterised by a series of shelves arranged around the line area (represented by red squares in the layout representation), as shown in Figure 4.

Material Flow Analysis

Figure 4 shows the material flow in the areas of interest.

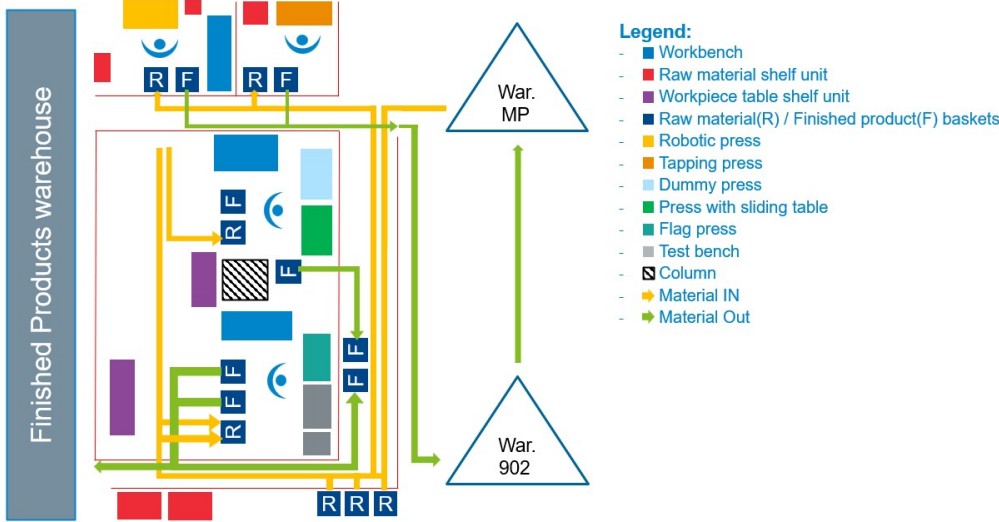

**Figure 4.** Graphic representation for the layout of stations involved in the project and the relative material flow.

Raw materials that had to reach the pre-assembly areas F301 and F306 were taken from the raw materials warehouse and brought to the respective lines in the areas marked with an M. These areas were not identified by horizontal signs but were the places where operators usually placed baskets of raw materials. If the raw material handling areas were occupied, the baskets of raw materials were momentarily stacked along the corridor leading to the finished products warehouse (in the schematic diagram, the lower right area). It should be noted that these movements were not always conducted by the logistic operators: often, the assembly operators were involved in those kinds of activities.

Once the material had been processed in the two pre-assembly areas, it was moved to the area of the finished product, marked with an F (also not standardised). Sometimes, to free up space, if the material was not promptly moved by the logistics department, the assembly operators independently chose to move the baskets along the passage corridors of the line (see F in front of the two assembly areas of FIS9).

Baskets of pre-assembled material were taken from areas F301 and F306 and then brought back to the warehouse at the 902 temporary location. The material remained there temporarily for 1 to 2 days and then were placed back in the raw material warehouse.

Once the final assembly order was launched, the pre-assembled items stored in the warehouse were taken from the raw materials warehouse and brought to the FIS9 assembly areas (with all the cases already described above). There, once processed, they were placed in the baskets of finished products and moved to the finished products warehouse.

Spaghetti Chart

The spaghetti chart graphically highlights the typical movements of operators in the shop floor. The result of the drafting of the spaghetti chart for the line in question is presented in Figure 5 (note that this kind of chart is typically hand-drawn).

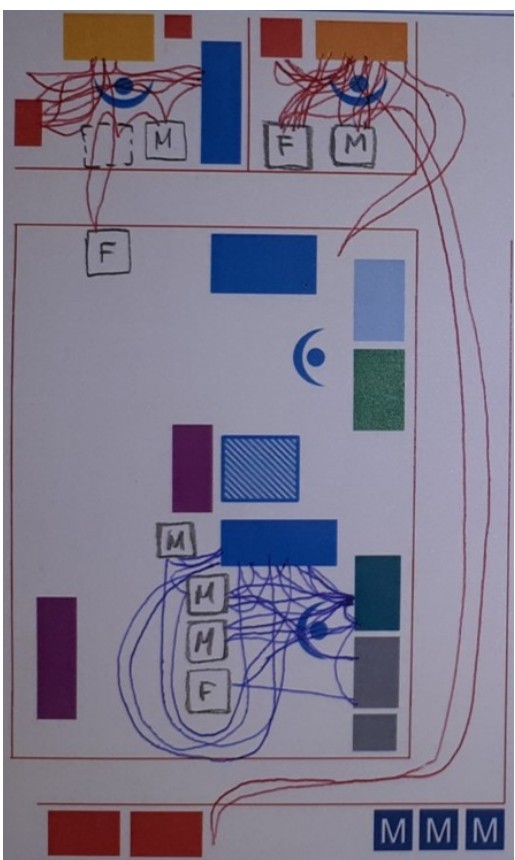

**Figure 5.** Spaghetti chart (typically hand-drawn).

Red lines represent the movements performed by the pre-assembly operators, whereas blue lines are the movements of the operators involved in the final assembly. Only the movements on the FIS9-B side were analysed. The results of the analysis are as follows:

- The operator of the F301 station takes about 60 steps during the analysis, resulting in a total length of 45 m.
- The operator of the F306 area takes 103 steps, resulting in a total distance of 77.25 m.
- The operator of the FIS9-B area takes 75 steps, resulting in a total distance of 56.25 m.

All those movements are considered as extra work, since they are mostly connected to the preparation of the area with the necessary materials needed to start a new work order, such as empty baskets to stock the finished product or raw material retrieved from the feeding structures scattered around the workplace. The ideal scenario is where the assembly operator has everything ready for the work to start, with minimum time spent for the machine setup (if any) and all the raw material promptly fed by the logistic department with a rational logic. For that reason, this document highlights the existence of useless movement, due to a low standardisation in the process, as reported in Section 2.1.1.

Muda Analysis

The word *Muda* is derived from Japanese: it means *waste*. Muda refers to waste of materials, work, and time. Seven types of Muda exist [2]: defects, overproduction, transport, waits, useless stocks, useless movement, and useless or expensive processes.

To analyse this process, specialised software was used (https://www.avix.eu/process-mapping-tools/time-and-motion-software accessed on 5 October 2022). The software took as input the video stream of the process, then gave the possibility to identify micro-actions within the process and to classify them using video cutting tools. The first software output quantified the amount of waste and its impact on the overall available work time based on the analysis made.

Video cuts representing micro-actions were interchangeable or disposable with respect to the kind of action the user wished to perform. This kind of feature has been used to project the "to-be" state associating every corrective action to be performed on the shop floor with the respective gain in efficiency, time, and cost. Further information is given in Section 2.1.2.

The high process variability dictated by the low standardisation was instead addressed through interviews on the shop floor. In this step, the experience of line operators and their leaders was taken into account. The result of the analysis is as follows:

- Total cycle time: 658.8 s/pc.
- Non-value added operations (NVA): 47%.
- Semi-value added operations (SVA): 26%.
- Value added operations (VA): 27%.

Analysis results are reported in Figure 6.

The working times identified by the analysis were divided into the three production areas according to the chart depicted in Figure 7.

The "tower" chart, or Yamazumi, highlights how VA, SVA, and NVA operations were distributed in the three areas under consideration. It is worth noting that island F301 highlights the time required for the pre-assembly of both crankcases and covers, while the FIS9 area is distinguished by two "towers" that represent the two stations working in parallel as if they were two separate areas, so the product does not pass through both—in other words, the products are assembled either in the first island or in the second. The cycle data were collected according to the reference model product. Of the seven Muda listed above, four emerged in the analysis:

- Overproduction, represented by two pre-assemblies; these were never launched as a result of the customer's order, but the output materials were produced "for the warehouse", thus representing a waste of space, resources, and capital immobilisation.

- Useless stocks: The pre-assembled items were stored waiting for the customer order, thus occupying space; even the related raw materials were not ordered and stored "when needed", but were actually immobilised "for the warehouse".
- Useless movements: The materials were moved around the plant twice more than needed, leading to delays and increased risks of damaging the material itself.
- Useless or expensive processes: This waste refers to the operations carried out by assembly operators. In fact, they not only carried out the assembly operations, but used to move baskets and crates to prepare their working areas; in addition, they took care of the material unpacking, waste management, and missed material retrieval.

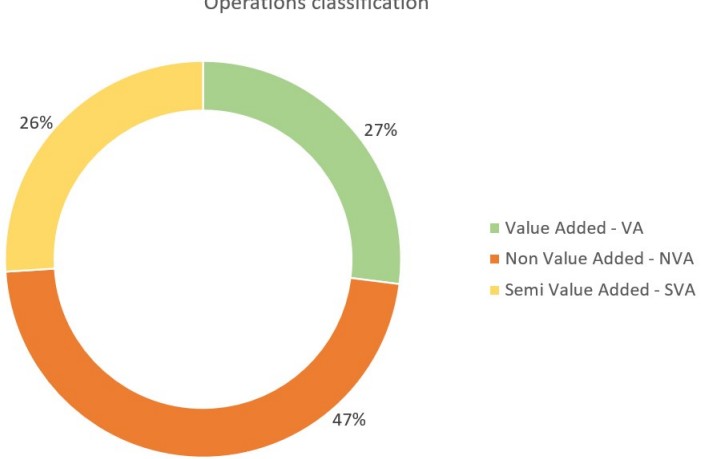

**Figure 6.** Operation classification in NVA, SVA, and VA of all the areas under examination.

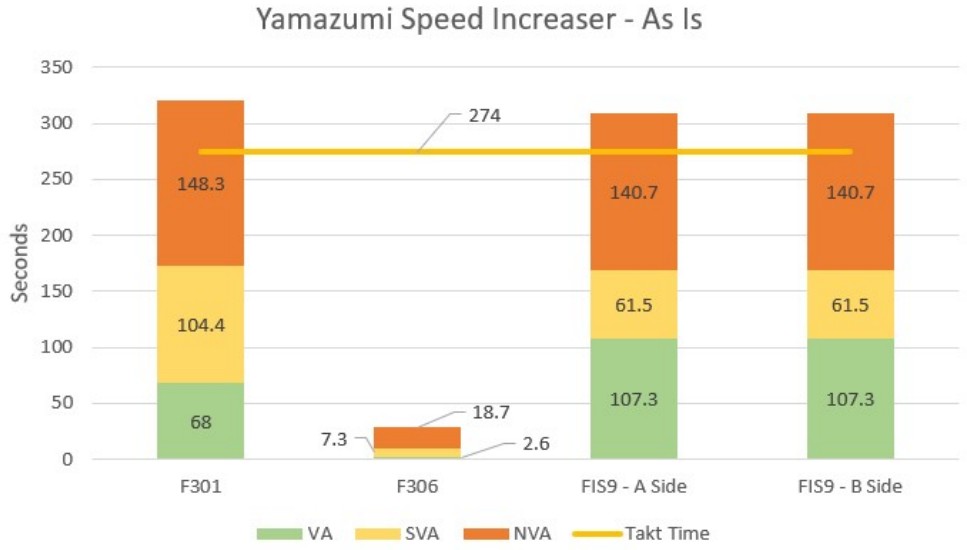

**Figure 7.** Yamazumi graph representing the impact, in seconds, of VA, SVA, and NVA operations on the total assembly time for every area.

Limits of Action

Additional constraints to the project were imposed by management:

- Maximising the reuse of the existing equipment.
- Freeing at minimum one of the three working areas involved in the "as-is" state.
- Implementing a solution with an investment payback not larger than 1 year.

2.1.2. The Proposed "To-Be" Solution

We now describe the proposed solution, focusing on the market requests, forecasts, and Takt time, as well as on the designing actions.

Market Requests, Forecasts, and Takt Time

The sales forecast for the year in which the analysis was performed (2019) indicates a request of 22,515 pieces. In the previous year (2018), the request was for 20,216 units. The data were provided by the management of the Operations Area, which interfaces directly with the company controllers. The product mix has not been calculated.

Based on the provided input data, we could compute the market Takt time for the current year. The Takt time is the rate at which the market requires the exit of a finished piece from the assembly line. This is a key information in the division and balance of assembly operations, as well as in defining the number of operators necessary to meet the request. The considered input data are as follows:

- Market request: 22,515 pieces/year.
- Daily hours available: 8 h/day.
- Work shifts: 1 shift/day.
- Available days per year: 230 days/year.
- Minimum physiological break: 7% (4% physiological, 3% fatigue - based on internal standards).

The market Takt time is computed in a single shift to maintain leeway for any peaks in requests; finally, this choice is useful for recovering the hours lost deriving from the project activity. With the given input data, a Takt time of 274 s per piece is obtained.

In the following section, we describe the designed actions to eliminate the detected waste and to standardise processes.

Designing Actions: Attacking Useless Movements and Useless Processes

The attack on these Muda is defined through four reported actions, which we describe in detail below.

- Action 1: The logistics workers must organise the raw material in crates. With this standardisation, operators do not have to open the packaging of raw materials, which are immediately available for assembly. Estimated time saving: 157.2 s per piece.
- Action 2: The definition of standard areas for positioning finished containers and raw materials. In this way, the operator does not have to move outside his area to prepare the finished container or recover that of the raw material. Furthermore, the definition of the areas simplifies the feeding of raw material and withdrawal of the finished product, favouring the coordination of operations between departments. Estimated time saving: 26.1 s per piece.
- Action 3: The introduction of a line feeding system based on customised roller conveyors, trolleys, and boxes. In this way, a unique interface is created between the logistics department and the assembly department: this enables the possibility to integrate a reordering kanban-based system that standardises the line supply signals with a consequent ease of coordination between departments. Finally, this action leads the island to operate in all respects with the lean method, abandoning the traditional work strategy. Estimated time saving of 110.1 s per piece.
- Action 4: Better positioning and management of assembly instructions. This action involves the adoption of reading desks (first) and computers (later), avoiding operators constantly moving from the workplace to search for the information necessary to complete the assembly activities. Estimated time savings of 11.2 s per piece.

A graph representing the impact of each action on the total of the detected Muda is reported in Figure 8.

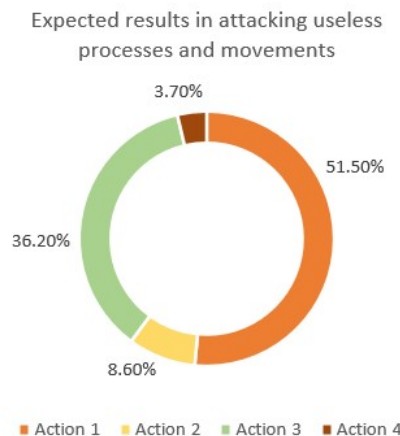

Expected results in attacking useless processes and movements

**Figure 8.** Expected impact of the four designed actions.

The actions described above lead to a reduction of the estimated cycle time from 658.8 s per piece to 354.2 s per piece, with a total saving of 304.6 s per piece.

Designing Actions: Attacking Useless Warehouses and Overproduction

Waste related to useless warehouses and overproduction mainly impacts the planning office and the logistics department. Here, the main change in the proposed solution was the elimination of pre-assemblies, which had no reason to be applied at this point in the supply chain due to the distance from the decoupling point and the Assembly-to-Order nature of the plant. As a result, the layout drastically changes, as well as the material flow. Figure 9 reports the change, as well as the material disposition with respect to the process balancing (see later).

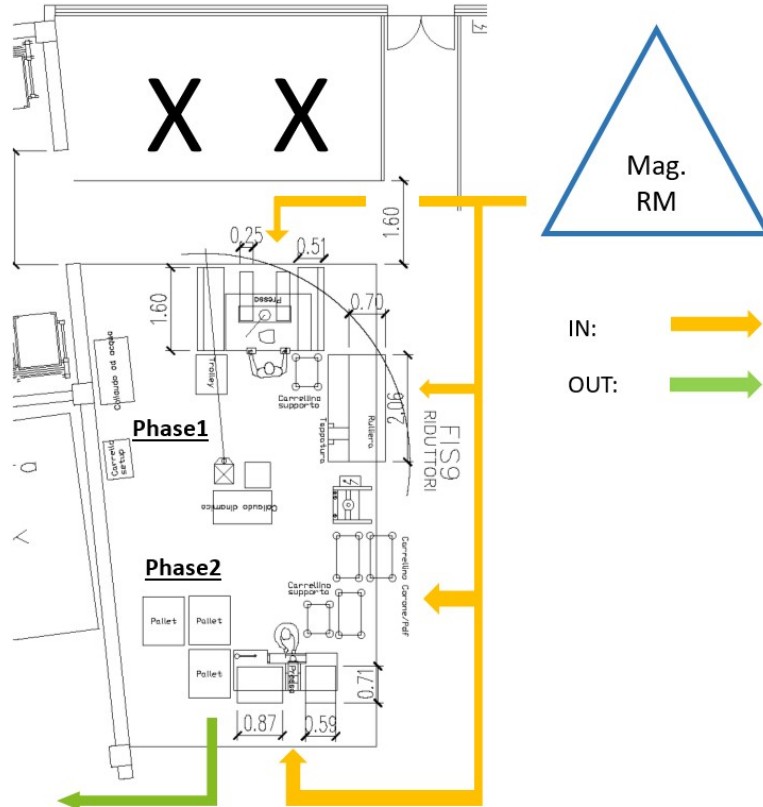

**Figure 9.** Updated layout with expected feeding flows.

Figure 10 shows the transformation of the office process, where the first two steps involving order production (ODP) and order transfer (OT) for pre-assembly (left) have become unnecessary due to the elimination of pre-assemblies (right).

The time saved at the planning office is summarised in Table 2.

**Table 2.** Main activities connected to the launch of a production order from the perspective of the planning office.

| | |
|---|---|
| Conversion from planning to ODP | 1.30 min |
| ODP release | 1 min |
| ODP printing | 2 min |
| OT creation + OT printing | 1 min |
| Time for ODP and OT | 5.3 min/(ODP + OT) |
| Number of ODP and OT launched in the last year | 590 (ODP + OT) |
| Total time spent in the last year | 5.3 min/(ODP + OT) * 590 (ODP + OT) = 3127 min |

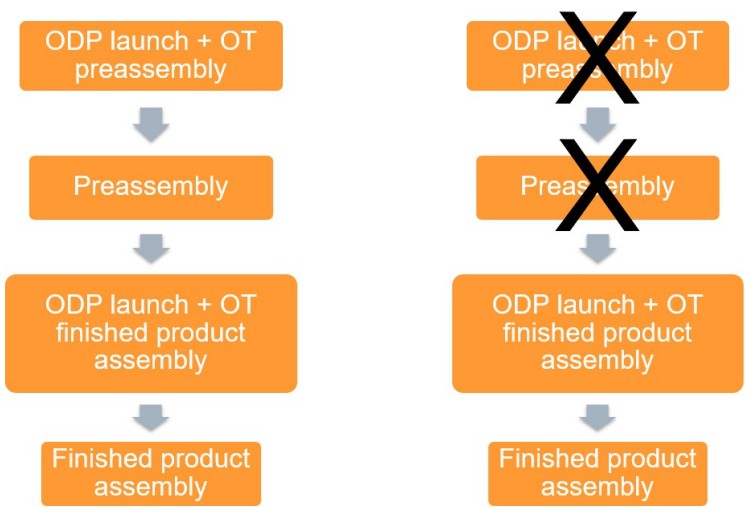

**Figure 10.** Office process with ("as-is", **left**) and without ("to-be", **right**) the pre-assemblies.

Similarly, Figure 11 shows the revised logistics process, where the unnecessary activities are those of pre-assembly, feeding of raw material, and withdrawal of finished subgroups.

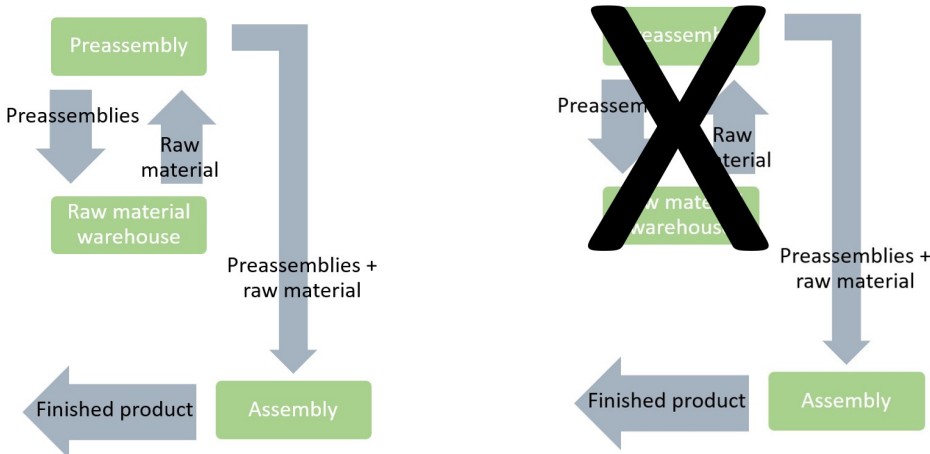

**Figure 11.** Logistics process with ("as-is", **left**) and without ("to-be", **right**) the pre-assemblies.

The saved times in the logistic operations area are summarised in Table 3.

**Table 3.** Main activities connected to the launch of a production order from a logistic perspective.

| | |
|---|---|
| Average warehouse/line/warehouse crossing time for raw material handling and pre-assembly | 2.58 min/(ODP + OT) |
| Total average time for line supply and pre-assembled withdrawal in the last year | 1522.22 min |
| Average withdrawal time for each OT line | 7 min/line |
| Total number of pre-assembly picking lines in the last year | 6,742 lines |
| Average pick-up time pre-assembled | 47,194 min |
| Total time spent in the last year for logistical management of pre-assembly | 51,843.2 min = 864 h |

As the focus of this paper is on the production phase, the aforementioned savings have not been considered in the subsequent discussion, since they fall under the logistics pillar. However, we decided to report them for completeness. The results described above and the actions related to them have been assigned to the logistics area, which has separately managed the issues and proposed improvements.

Designing Actions: Process Definition and Balancing

In principle, the elimination of Muda would result in a total cycle time of 354.2 s per piece. However, this is greater than the Takt time required by the market (274 s per piece). Therefore, two options have been considered: (1) assembling the product in two work shifts, with at least with one operator per shift; (2) assembling the product in one shift with the use of two operators. From the point of view of labour costs and required time, the two options are identical. We thus chose the second one, in order to model an assembly line that can absorb any peak demand. On the other hand, in the event that demand fell, the use of two operators at the same time would result in a new waste: to prevent it, the line will be "U-shaped" to favour its functionality even with only one operator. The result of the line balancing process is summarised in Figure 12.

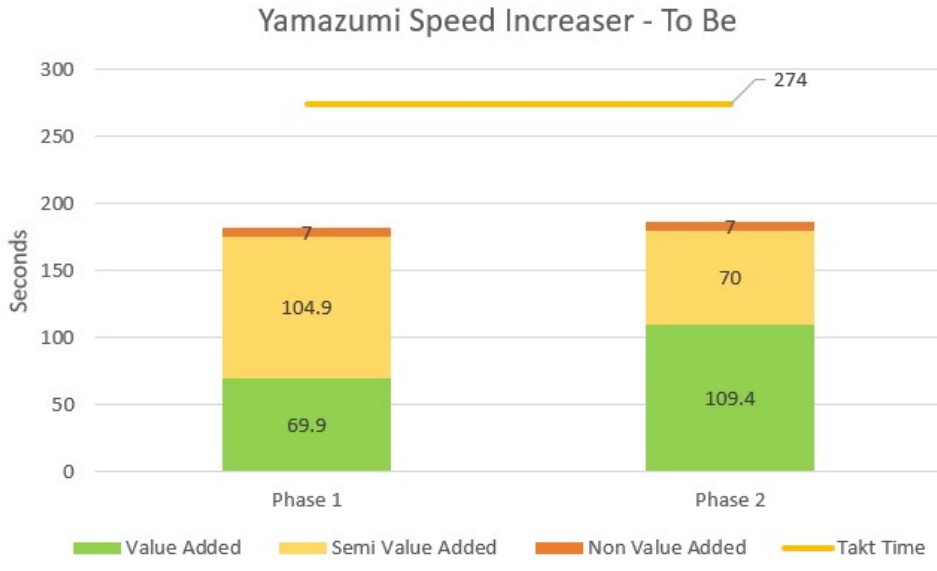

**Figure 12.** Yamazumi chart for the "to-be" expected state: it is worth noting the result of the operations balancing, in which 7 s have been added for each phase to take into account the steps that the operator must perform to move along the line (estimated at about 1 second per step).

Operations in Phase 1 include the greasing of bearing seats in crankcases and covers, the assembly of bearings and oil seals in crankcases and covers at the same time, the preparation and assembly of oil plugs for crankcases and lids at the same time, and pinion capping. Operations in Phase 2 include inserting dowel pins, gasket and glue for gaskets; preparation and insertion of the power take-off in the crown; insertion of the power take-off and pinion in the crankcase; and multiplier closure with lid and pneumatic test.

### 2.2. Oil Seal Control with Deep Learning

During the process of the analysis of the assembly line, a frequent quality problem emerged: the leakage of the speed increaser around the shaft. Most of the time, the issue was discovered as a customer claim—after the customer had bought the speed increaser and tried it directly in the field. Despite the problem not being directly involved in the productivity improvement project, it affects the First Time Quality index of the assembly line, lowering its final overall performance. We devised a solution to be easily plugged in the assembly line, based on Poka Yoke and deep learning.

#### 2.2.1. Problem Description

The problem appeared indiscriminately for any final product. The inspections made after the customer return highlighted that all cases were generated because the oil sealing had been mounted upside down.

The inspection process was performed using compressed air. Basically, a compressor was mounted at the oil load seat, and air was pushed inside the speed increaser at 3 to 5 bar. Then, a control unit monitored the test cycle and, in case the requirements were met, the product was validated. The air test does not in any case reproduce exactly the presence of oil. Therefore, spotting a leakage generated by an inverted seal is not guaranteed. Instead, spotting a missed seal in these conditions is always guaranteed.

A possible solution could emerge by adding a different testing gas to the receipt (e.g., nitrogen or helium). In this scenario, a higher level of leakage sensitivity could be obtained. In any case, this solution only partially solves the problem, because of the position of the test within the assembly line. In fact, the leakage test has to be performed at the end of the line, despite the sealing mounting being performed far before it (in order to perform a leakage test, it is necessary for the product to be "closed", which happens only in the very final stage of the line). In case of the identification of an incorrectly placed seal, the problem could only be fixed with the complete disassembly of the piece, as the oil seal is mounted under the bearing and is entirely surrounded by the bearing itself and the crankcase/cover (in Figure 13, the oil seal and bearing housing are presented). The bearing extraction, in particular, is a critical phase, because it is coupled with interference in its seat. Therefore, to extract it, specific tools are needed (extractors), a high use of force is required, and a long time is needed as well. For those reasons, even the ability to spot the wrong piece at the end of the line is not enough, because the scenario would still require the use of a dedicated area and of a high-skilled operator.

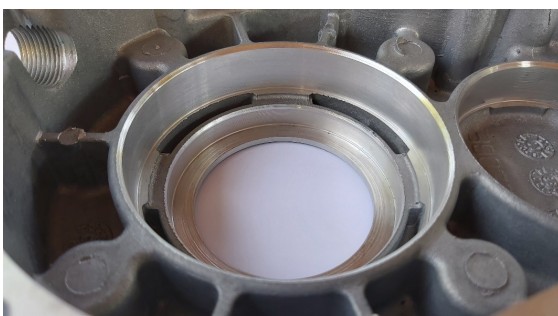

**Figure 13.** Oil seal and bearing housing.

To completely eliminate the problem, it is therefore necessary to implement a monitoring system in the place where the oil seal and the relative bearings are mounted. Nevertheless, implementing a quality check test or a traditional Poka Yoke might be challenging and costly, because of the nature of the involved pieces. Moreover, no process solution had been found by the company so far, and even a product re-design (which is also a costly option) could not guarantee its effectiveness. For all those reasons, a solution that overcomes the traditional lean tools and that can seamlessly integrate with them is needed.

### 2.2.2. Proposed Framework

The diameters of the oil seals vary between approximately 50 and 70 millimetres, depending on the model in which they are used. Given the small size of the oil seal, together with the uniformly black colour, recognising one side or the other turns out to be an activity that requires a high focus by the operator: with an insufficient light for the inspection, and with many working hours behind (resulting in an accumulation of mental fatigue), it is easy to commit mistakes (Figure 14).

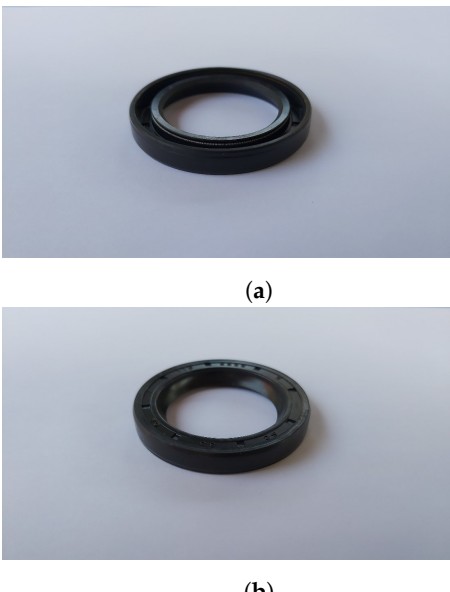

(**a**)

(**b**)

**Figure 14.** The two side of oil seals. (**a**) Oil seal side that has to face the inside of the product. (**b**) Oil seal side that has to face its crankcase housing.

The challenge lies in implementing a method of verifying the proper positioning of the oil seal before the relative bearing insertion, while the operator is performing the assembly operation. The reason is that, in the event that there is no simultaneity between the control and the assembly, there is a risk that the operator inadvertently positions the oil seal upside down (or even forgets it), making the control process itself useless.

The proposed solution consists of a camera, fixed in the lower part of the press arm, which has to inspect the whole process and to detect possible mistakes. A mock-up of the solution is depicted in Figure 15.

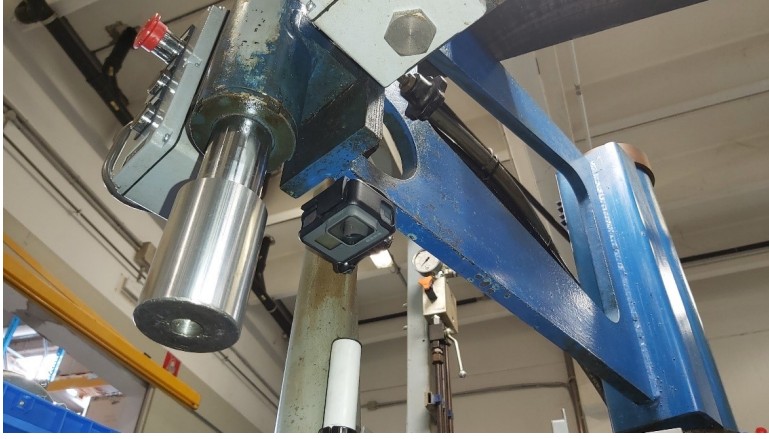

**Figure 15.** Camera mock-up showing the point of view.

The recognition system must be able to detect the components even if they are not positioned exactly in the same way at each cycle. This is necessary because the oil seal can assume slightly different positions before the normal activation of the press: in particular, in addition to being positioned straight or upside down, during straight positioning, the lower face of the oil seal may not be perfectly parallel with respect to the support surface of the crankcase, resulting in a "crooked" and unpredictable direction (detail reported in Figure 16). On the process side, this is not a crucial problem, due to the rubber nature of the oil seal: it is sufficient for it to be positioned in the surroundings of its seat to be placed correctly using the press. Nevertheless, the monitoring unit has to be able to discern this characteristic, without losing the ability to recognise the side of the oil seal in place.

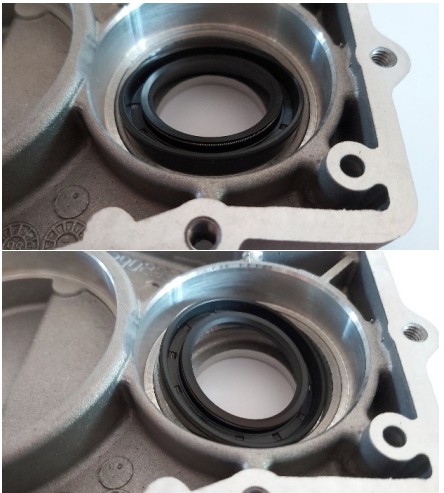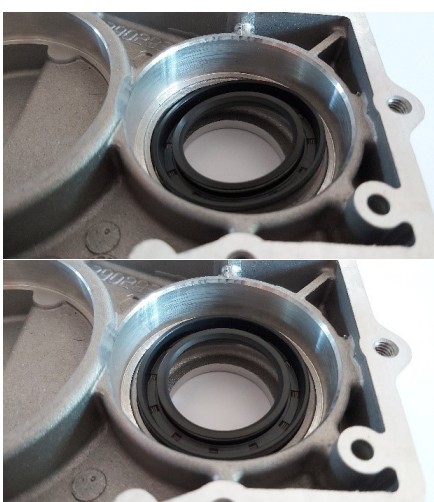

**Figure 16.** Possible positioning of the oil seal above its house before the use of the hydraulic press.

Finally, the system must recognise the components from different angles, as the camera will move along with the head of the press along an arc of circumference (see Figure 17). The descent process of the press head would take place only if the camera detects the correct pre-positioning of the oil seal in its place. A diagram of the resulting process is depicted in Figure 18.

The schema highlights how the control unit allows the activation of the press to recognise the correct positioning of the oil seal, fulfilling in all aspects the role of a direct controller of the process, preventing in any case the error in a full Poka Yoke spirit. This solution is not only easy to integrate into the process but ensures the complete and easy scalability of the inspection for all the speed increaser models assembled here.

Among the other errors often made by the operator due to the dated engineering of the press drives, there is the incorrect selection of the driving force of the bearings and oil seals: in fact, the press offers the possibility to carry out the operation at 50 kg, 750 kg, and 1250 kg; the oil seals are planted with 50 kg of force, with the bearings planted with 750 kg of force. The incorrect selection of the planting force does not represent a quality risk; however, once the solution proposed above has been implemented, it would be very simple to automatically select the working pressure according to the component positioned on the crankcase/lid.

### 2.2.3. Methods

The problem described above can be addressed as a classification task: given the image of an oil seal, the goal is to recognise whether the object (1) is in the correct position, (2) has been positioned backwards, or (3) is missing. In this work, we use a state-of-the-art algorithm for image classification, namely convolutional neural networks (CNNs).

CNNs are artificial neural networks equipped with the so-called convolutional filters, whose application leads to an efficient recognition of the images by the machine during the learning phase. Those filters aim to automatically extract useful information (i.e.,

features) from images, such as horizontal lines, vertical lines, angles, or even more complex patterns. The filtered image is called a feature map. This kind of network has proven extremely powerful for several computer vision tasks, now becoming a cornerstone of most state-of-the-art applications [34].

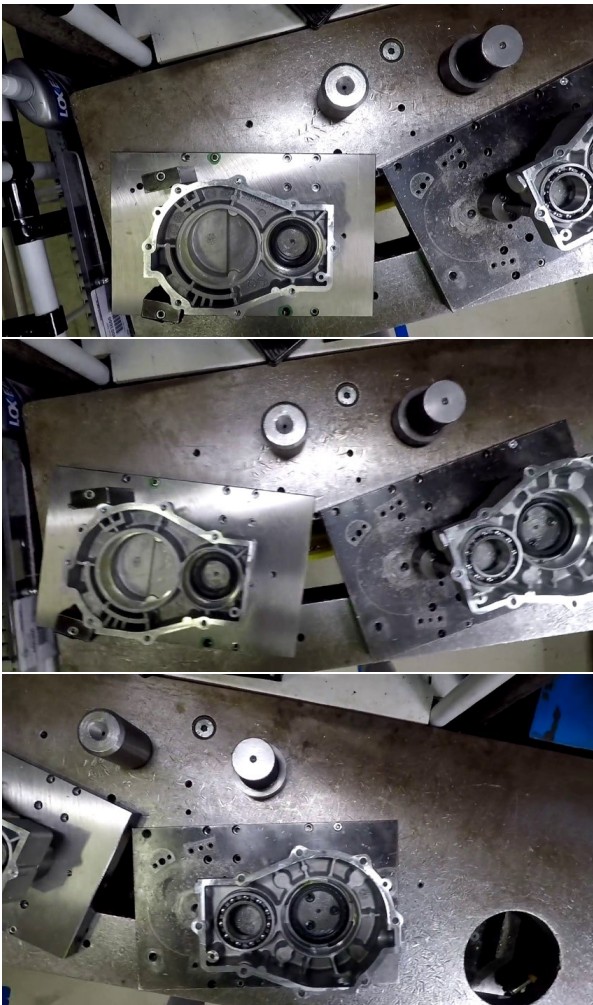

**Figure 17.** Camera point of view.

2.2.4. Dataset

In order to train the CNN, 500 full HD format images (1920 × 1080 pixels) were collected through the use of a common mid-range smartphone and a tripod. The pictures depict a crankcase and an oil seal. The chosen model was the 12610004—the model product involved in the previous assembly line re-design. We collected samples for the three possible oil seal configurations (not positioned, positioned upside-down, correctly positioned) that we want the system to learn and recognise. Figure 19 shows an example of an image collected for each class. It is worth noting the difficulty in recognising the difference between a correctly positioned oil seal and an incorrectly positioned one—the key point of the project. All images were resized to 240 × 320 pixels and converted to grey-scale (since the colour does not bring meaningful information) to reduce the computational burden. A standard pre-processing phase with a 5 × 5 high-pass filter was also applied in order to highlight borders and other high-frequency information that is crucial for the classification task.

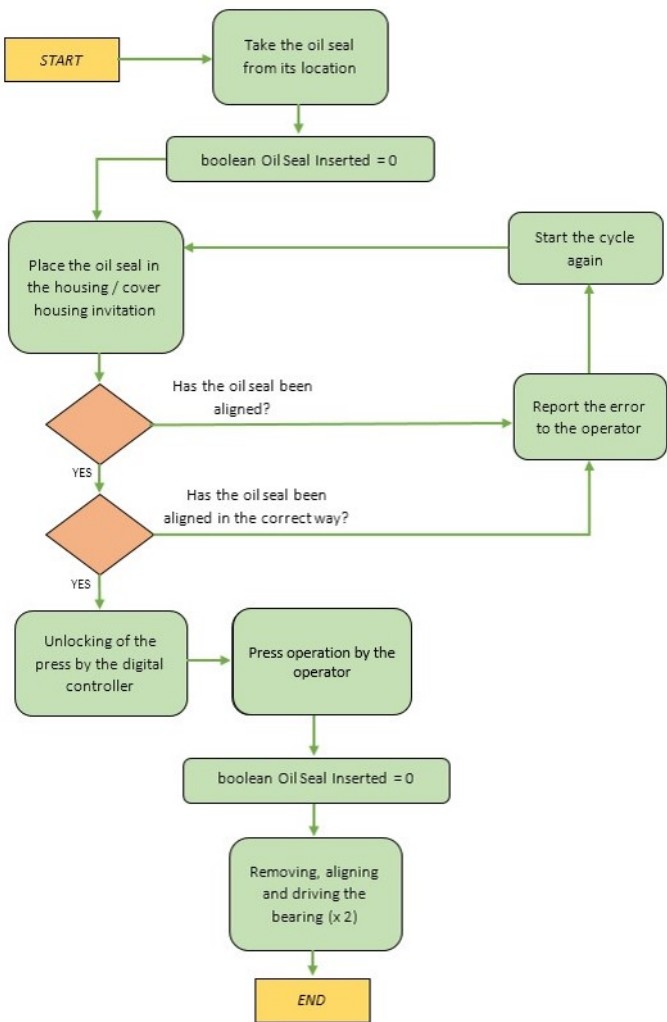

**Figure 18.** Operational flow diagram.

2.2.5. Experiments

In our experiments, we aimed to trade-off accuracy and model complexity in order to obtain a solution that does not require large computational resources and could be easily embedded within an on-board system. We used a CNN with a single convolutional layer consisting of 32 filters with dimension $3 \times 3$ and rectifier activation function. We then applied a max pooling layer with size $2 \times 2$. After flattening the output, we used a dense layer with 64 neurons, a rectifier activation function, a dropout layer with a probability of 0.5, and finally an output layer with 3 neurons to perform multi-class classification.

The Adam optimiser was used to train the network, using categorical cross-entropy as the loss function, as is customary in multi-class problems. During training, we used mini-batches of size 32 and performed early stopping on the validation loss with a patience equal to 15 epochs. The training phase took place in the cloud through the use of the GPUs offered by the Google Colab service (https://colab.research.google.com/ accessed on 5 October 2022). Note that the hyper-parameters defined so far are the result of a trial and error phase, where their combination provided the best result on the validation set. An explanatory loss graph is reported in Figure 20. An early stop occurred at the 30th epoch over 500.

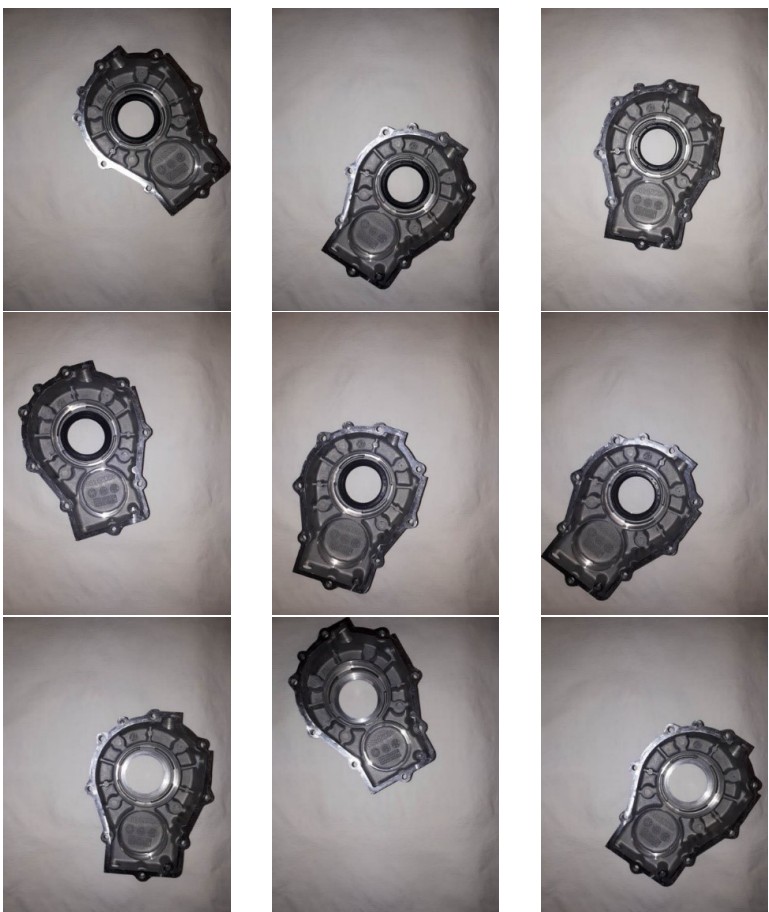

**Figure 19.** Example images of the data collected. The first three pictures represent the correct position of the oil seal, the second three the oil seal positioned upside-down, and the last three is the absent oil seal.

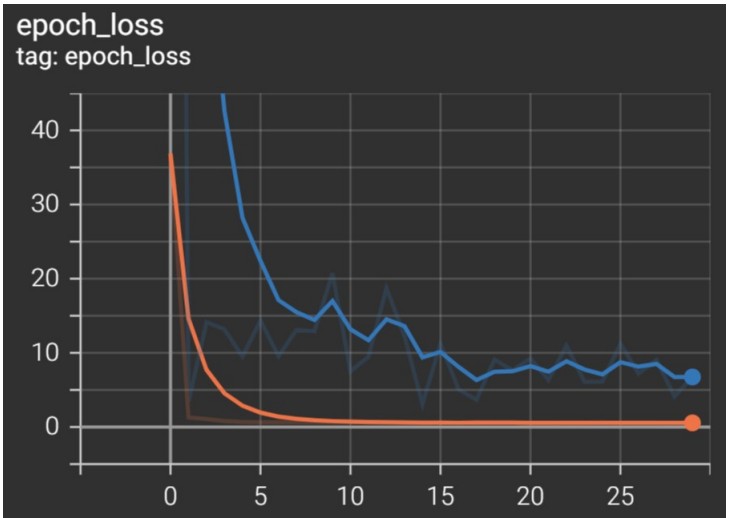

**Figure 20.** Graph of the loss trend with respect to epochs: the train loss is reported in red, while the validation is in blue. A smooth factor of 0.6 has been applied.

For the evaluation of our approach, we performed a repeated *k*-fold cross-validation. In a *k*-fold cross-validation, the original data set is partitioned into *k* sets: in turn, each set acts as the test set, while the remaining $k - 1$ sets are used for training and validation. Results can be aggregated either via the macro-average or micro-average. To make the

procedure robust, the *k*-fold cross-validation can be repeated *n* times, and average performance measures are usually reported. In our case, the number of folds *k* was set equal to 5.

The average micro-accuracy on the five folds is equal to 98.8%, ranging from 97.0% up to 99.1%. Given the small amount of images used in the training set, the performance of the CNN is indeed very encouraging, as the model is capable of recognising the different orientations of the target object even if the details to consider are very small.

## 3. Results

### 3.1. Assembly Line Improvement Project

We hereby summarise the results of our analysis by highlighting the overall savings that the re-design of the assembly line has produced. In the "as-is" scenario, we have the following initial situation:

- Cycle time: 658.8 s/pc.
- Real hours spent in the last year: 5654 h.
- Real hours spent in the last year for assembly activities only: 5019 h.
- Hours per piece allocated in non-assembly activities: $(5654 − 5019)$ h/20,216 pc = 113.1 s/pc.

The last item was computed as an indicator of the overall productivity, i.e., considering both the time dedicated to direct and indirect activities of the line, assuming that the indirect activities do not change.

The "to-be" scenario gives instead the following results:

- Cycle time (assembly Takt time): 186.4 s/pc.
- Total time spent for assembling a piece: 186.4 s/pc * 2 operators = 372.8 s/pc.
- Saving per piece: (658.8 – 372.8) s/pc = 286.0 s/pc.
- Total saving in current year: 286 s/pc * 22,515 pc = 1789 h,
- Efficiency: 348 s/pc/372.8 s/pc = 93%.
- Productivity: 348 s/pc/(372.8 s/pc + 113.1 s/pc) = 72%

Additionally, 864 hours were saved in the logistics department and programming office, as the management of the pre-assembled components was no longer necessary.

The total cost for implementing the proposed solution can be estimated as follows:

- Roller conveyors, interlocking structures complete with tool racks, structures for electronic screwdriver control units, and process support trolleys: 8000.
- Renewed piece holder boards for BF10: 5000 EUR.
- Layout modification, modifications to the electrical system and safe positioning of the manipulator: 3000 EUR.

The total cost of implementing the solution is thus 16,000 EUR. The estimate of cost benefits is therefore proposed below:

$$B/C = (1789 \text{ h} * 0.8 * 30 \text{ EUR/h})/16{,}000 \text{ EUR} = 2.68$$

Such an estimate was made by assuming to be able to reach 80% of the estimated result, as a precautionary measure. In addition, the hourly cost of operators was valued at 30 EUR per hour, as per company practice. The result indicates that the project will have a positive impact on the assembly line, with a payback time of just over 6 months.

### 3.2. Intelligent Poka Yoke Metrics and Expected Benefits

To better analyse the errors of our model, we show in Table 4 the resulting confusion matrix, micro-averaged over the five folds in the cross-validation. The matrix shows that the CNN never classifies an oil seal that is wrongly positioned as correctly positioned. This is extremely important in our industrial scenario, since such an erroneous classification would enable the use of the hydraulic press, thus generating the kind of waste that the system is trying to avoid.

**Table 4.** Confusion matrix of the classification task, micro-averaged over the five folds.

|  | Correct | Reverse | Absent |
| --- | --- | --- | --- |
| Correct | 146 | 0 | 4 |
| Reverse | 0 | 150 | 0 |
| Absent | 2 | 0 | 199 |

Instead, an oil seal that is correctly positioned classified as incorrectly positioned will not generate quality issues: the hydraulic press will thus not be enabled, and the only action the operator will have to perform is a *new classification*, moving away the camera and reframing the object. This characteristic highlights the high reliability of the solution, which will positively impact on the quality production metrics and customer quality perception of the assembly system for that particular kind of waste.

The experiment is a clear example of how AI applied to industry can help in reaching the zero-defect target, thus making the technology a *six-sigma* practical tool. Six-sigma can be defined as "a problem-solving method that uses quality and statistical tools for basic process improvements" [35], whose target is to reduce variability and waste in process in order to have total control over the process.By looking at the confusion matrix, the high precision for the class representing the incorrectly positioned oil seal allows us to argue that the result reflects (and reaches) the six-sigma goal. The overall savings have been computed taking into account the reported project implementation cost with respect to an alternative project, which involved the engineering department redesigning the crankcase and the lid in order to avoid the analysed problem. The common target of both scenarios is to completely eliminate the studied type of waste, which justifies the kind of analysis chosen. In the following, the redesign expected cost structure is described:

- Number of hours needed for new crankcases and lids designs, according to given requirements: between 960 and 1920 h.
- Average cost of the engineering department: 50 EUR/h.
- Expected redesign cost: [50 EUR/h * 960 h = 48,000 EUR; 50 EUR/h * 1920 h = 96,000 EUR].
- Average expected redesign cost: 72,000 EUR.
- New moulds cost: not considered.
- Tests cost: not considered.

Therefore, the expected cost for a complete redesign of all the products involved in the issue is around 72,000 EUR. The described project cost structure is presented below:

- Expected hours needed for data acquisition, implementation and validation: 480 h;
- Average cost of the engineering department: 50 EUR/h.
- Expected redesign cost: 50 EUR/h * 480 h = 24,000 EUR.
- Camera vision system: 2000 EUR.
- industrial computer: 1000 EUR.
- Additional electrical equipment: 200 EUR.
- Expected installation hours: 4 h.
- Average cost of maintenance department: 30 EUR/h.
- Expected installation cost: 30 EUR/h * 4 h = 120 €.

The total expected cost of the presented project is around 27,320 EUR, which is less than half of the cost of a complete crankcase and lid redesign.

*Generalisation* is helpful also in the first system implementation, as it means that it is not necessary to have all the involved products represented in the data set: only a subset of them is required, which is identified in the most common assembled product (see the Section 2.1.1 for reference). In fact, different speed increaser models do not differ that much in terms of their geometry structure, as well as the ball bearings' oil seals. Finally, the generalisation gives robustness and flexibility to the system at the introduction of new products: in fact, their introduction is not necessarily expected to result in the necessity for new model training and validation. Nevertheless, if the trained system highlights a

difficulty in having good classification performances for new product, operating a new data acquisition, training, and deployment phase would be far more agile with respect to gaining the same error-proof feature during the design phase.

The deployment phase could involve a new data collection as well. In addition, the generalisation capabilities of CNN can be exploited as well, as previously described. For that reason, a preliminary test with the trained CNN is also encouraged.

Then, the use of an industrial computer connected to a camera positioned as depicted in Figure 15 is necessary. The computer has to be also connected to the press electric circuit in order to make its functionalities available if and only if the oil seal is positioned correctly. The functionality can be achieved via the use of simple relays guided by a low-voltage circuit (e.g., 5 V circuit).

The proved solution can also be easily extended on the software side implementing an automatic force selector, considering an adequate integration also on the hardware side. In this way, the process would achieve a higher level of quality, avoiding scraps generated by the erroneous use of the pressing force.

### 4. Conclusions

In this work, we described the re-engineering process of an assembly line with Lean Manufacturing methods. The result of the process led to an increment of the line productivity from 46% to 80%. The re-engineering process addressed, among others, the problem of the incorrect assembly of oil seals in the final assembled product. This problem was particularly important in the assembly line as it affects the First Time Quality index of the area. Because of the nature of the problem, a novel solution approach was needed. We proposed the exploitation of a deep learning approach, namely a convolutional neural network, to automatically detect the incorrect positioning of the oil seal. The designed architecture resulted to be a Poka Yoke, thus representing an interesting application of modern technologies based on artificial intelligence directly on the shop floor. The presented case study can be seen as an example of synergies present between Lean Manufacturing tools, six-sigma, and Industry 4.0 [36,37]. Industry 4.0 pillars, in fact, enable the evolution of Lean Manufacturing and six-sigma concepts, leading to the so called "Lean Six-Sigma 4.0" [36], represented by [38] in the reversed cone model. We believe that similar applications of this kind of technique will be increasingly present in the manufacturing industry in the coming years, as a key component of Lean Manufacturing tools and methods.

**Author Contributions:** M.M.: conceptualisation, software, formal analysis, investigation, writing—original draft, writing—review and editing; M.L.: methodology, validation, resources, supervision; R.G.: methodology, supervision. All authors have read and agreed to the published version of the manuscript.

**Funding:** This research received no external funding.

**Institutional Review Board Statement:** Not applicable.

**Informed Consent Statement:** Not applicable.

**Data Availability Statement:** Not applicable.

**Conflicts of Interest:** The authors declare no conflict of interest.

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
