# Peer review of "Poka Yoke Meets Deep Learning: A Proof of Concept for an Assembly Line Application"

_applsci, doi:10.3390/app122111071_

Round 1

Reviewer 1 Report

I found the paper very interesting.  I think you did a good job

Author Response

Thank you very much for appreciating our work.

Reviewer 2 Report

·         A brief summary

To achieve long-term success, a continuous improvement, innovations and changes are imperatives for any industrial company at present.  That is why, the content of this paper ought to be considered as very topical! In spite of the paper is not strictly scientific treatise, it is a nice contribution to the world-wide discussions how to interconnect Industry 4.0, industrial engineering and Lean approaches or tools by effective and efficient manner. The main and inspiring results are included into parts 3 and 4. In total, quality of the paper is satisfactory.

·         Broad comments

Strengths:

-          Authors clearly describe two ways and key steps how to plan and implement activities in area of assembly line performance.

-          Solutions presented into parts 3 and 4 are very interesting and inspiring for all members of inherent community.

-          Although authors used different indicators (lines 428 – 445) for illustration of positive performance shifting, an estimated payback time of the assembly line redefining is fully respectable.

Weaknesses:

-          The structure of the paper mostly does not comply with MDPI requirements (see mdpi.com/authors). I recommend to adapt and newly distribute all text to following parts: Introduction, Materials and Methods, Results, Discussion, Conclusions, References. Text placed in lines from 52 to 56 should be the also adapted.

-          To increase a scientific value of the paper, authors should enrich the literature review by other examples or references from additional resources, especially focused on mutual linkage between Industry/Quality 4.0 and Lean concepts.

-          A cost – benefit analysis is missing in case of the second project, presented into part 4.

-          Authors did not mention year (or time span) of the project implementation as well as impacts on external customers perception.

·         Specific comments

-          Line 10: Terms “Golden zone” and “Strike zone” should be explained.

-          Table 1: Acronyms like DP, P/R, P/F/S, and HIL should be explained immediately within table title. This table should be placed near to the first time it is cited - it means below line 186.

-          Figures 3 as well as 4 should be enlarged, letters and numbers are difficult to read.

-          Figures 8, 11, 17 and 18 are not cited in the main text.

-          Figure 13: Does really the arrow “Preassemblies” occur within state “to-be” – see right scheme at the figure?

-          All figures in part 4 should be placed the nearest to the first time they are cited.

Authors should see all weaknesses as well as specific comments mentioned above as areas of possible paper´s refinement!

Author Response

We thank this reviewer for their constructive comments. Please find our reply to your comments below.  

The structure of the paper mostly does not comply with MDPI requirements (see mdpi.com/authors). I recommend to adapt and newly distribute all text to following parts: Introduction, Materials and Methods, Results, Discussion, Conclusions, References. Text placed in lines from 52 to 56 should be the also adapted.
>The paper structure has been reorganised as indicated by the reviewer.  

To increase a scientific value of the paper, authors should enrich the
literature review by other examples or references from additional re-
sources, especially focused on mutual linkage between Industry/Quality
4.0 and Lean concepts.
>Additional material referred to suggested topics has been added.  

A cost – benefit analysis is missing in case of the second project,
presented into part 4.
>A cost-benefit analysis has been introduced in Section 3.2. Since the project
was competing with a traditional way to solve the issue (a complete product
redesign for involved product models), a comparison between the two scenarios
has been done.  

Authors did not mention year (or time span) of the project implementation as well as impacts on external customers perception.
>The project time span has been clearly mentioned in the cost-benefit analysis
introduced in Section 3.2. The impact on customers has been discussed with a
brief note, since it is not the main focus of our work.  

Line 10: Terms “Golden zone” and “Strike zone” should be explained.
>The two terms have been explained in the abstract.  

Table 1: Acronyms like DP, P/R, P/F/S, and HIL should be explained immediately within table title. This table should be placed near to the first time it is cited - it means below line 186.
>Acronyms as well as table placement have been changed following this suggestion.  

Figures 3 as well as 4 should be enlarged, letters and numbers are
difficult to read.
>The two figures have been adjusted accordingly. The rightmost part of the
figure has been truncated in order to gain readability.  

Figures 8, 11, 17 and 18 are not cited in the main text.
>References have now been properly introduced.  

Figure 13: Does really the arrow “Preassemblies” occur within state
“to-be” – see right scheme at the figure?
>No, the arrow “Preassemblies” does not occur. All the steps involved in the
process described by the “Preassembly” and ”Raw material warehouse” boxes,
and the arrows between them, have been eliminated in the passage from the
“as-is” to the “to-be”. The image has been modified accordingly.  

All figures in part 4 should be placed the nearest to the first time they
are cited.
>Figures have been moved as near as possible to the place where they are
cited. It was not possible to rearrange all of them because of the size of some
(e.g., Figures 19, 20 and 21), but we believe that the arrangement now makes
more sense.  

Authors should see all weaknesses as well as specific comments men-
tioned above as areas of possible paper ́s refinement!
>Thank you very much for appreciating our work.

Reviewer 3 Report

My Comments and Suggestions for the Authors are given below:

1.     The authors present a case study of the re-engineering process of an assembly line with speed reducers and multipliers for agricultural applications. The presented case study can be seen as an interesting application example of the synergies of lean manufacturing tools, Six-Sigma and Industry 4.0, i.e., the study is a new generation of the Poka-Yoke.

2.     The abstract should be rewritten to better present the article content.

3.     A more comprehensive and systematic literature search should be carried out.

4.     The quality and visualization of all shapes should be improved. Fonts and other aspects should be standard.

5.     Figure titles should be kept shorter and explanations should be made within the paragraph (for example, Figure 21).

6.     The information given separately in Figures 5 and 6 can be combined and presented in a single figure.

7.     The labels in Figure 8 can be listed on the side instead of showing them at the bottom.

8.     How do image distortions in the camera affect the results of your CNN model in case of a press head shake?

9.     How many images did you use for each size of crankcases and oil seals for the CNN training?

10.  You have not included the training and loss graphs that show the performance of your CNN model.

11.  It is understood that the examples you collected for CNN training and the images obtained with the actual camera are different. In this case, image processing is required in the natural environment. Without this, your CNN model cannot understand the image to be classified. How did you get the image area you are interested in with image processing?

12.  You have not included the sample images you obtained with your deep learning model.

13.  Conditions such as ambient light conditions and size changes in the image adversely affect deep learning results. Did you use any data augmentation methods to avoid these situations?

14.  In the article, in addition to error prevention, it was expected to reduce non-value added operations further with improvement suggestions in processes. It is seen that this gap is low. Criticizing this, a comparative evaluation and analysis of the results obtained should be done collectively in a table.

15.  Spelling errors and other errors should be corrected.

16.  Conclusions should be rewritten: references given in the conclusions should be given in the previous sections. In conclusions, mainly the obtained results in the study should be presented, and the limitations of the study and future studies should be explained.

Author Response

We thank this reviewer for their constructive comments. Please find our reply to your comments below.
2. The abstract should be rewritten to better present the article content.
>The abstract has been re-written following this suggestion.  

3. A more comprehensive and systematic literature search should be carried out.
>Related works have now been included in the introduction, following the suggestion of Reviewer #2 to organise the paper using the typical sections used by MDPI. Moreover, additional material has been provided. Relevant literature is now described in several subsection of the introduction, according to a classification of the topics and contents.  

4. The quality and visualization of all shapes should be improved. Fonts and other aspects should be standard.
>Fonts has been check as well as the overall visualisation, in accordance also to hints given by reviewer #2.  

5. Figure titles should be kept shorter and explanations should be made within the paragraph (for example, Figure 21).
>Caption of Figure 21 has been shortened and part of it has been integrated
in the main text. The caption of Table 1 has been shortened as well, following
also the suggestions of Reviewer #2. All the other captions have been revised.  

6. The information given separately in Figures 5 and 6 can be combined and presented in a single figure.
>The two images have been combined as suggested.  

7. The labels in Figure 8 can be listed on the side instead of showing them at the bottom.
>Labels of Figure 8 have been changed accordingly.  

8. How do image distortions in the camera affect the results of your CNN model in case of a press head shake?
>The press head shake cannot occur when the recognition system is in action. In fact, first the camera performs the classification and only then the press is released. Then, the press is kept available for a limited amount of time, which is set with respect to the average time needed for the operator to put the piece in place. If the operator misses the time-window, he can perform a new classification in order to have the press newly available.  

9. How many images did you use for each size of crankcases and oil seals for the CNN training?
>500 images containing all the 3 classes have been used. Details are given in Section 2.2.4.  

10. You have not included the training and loss graphs that show the performance of your CNN model.
>An explanatory train and loss graph has been added as figure 21.  

11. It is understood that the examples you collected for CNN training and the images obtained with the actual camera are different. In this case, image processing is required in the natural environment. Without this, your CNN model cannot understand the image to be classified. How did you get the image area you are interested in with image processing?
>As explained in the paper, this project describes the prototype as a proof-of-
concept whose aim is to highlight the possibility to use Industry 4.0 technologies (such as AI) to address Lean Manufacturing problems. The computer vision system was evaluated offline, to assess its performance and impact on the overall system. The training and validation data set was therefore collected with this specific aim. For the deployment of the application, an on site/on line data collection process should be carried out, during normal working hours, with the selected camera already mounted on the press. In fact, the camera presence does not impact on everyday activities, giving no trouble to operators.  

12. You have not included the sample images you obtained with your deep learning model.
>Our deep learning model simply performs a classification task, thus it does
not produce any new images. Maybe we did not understand what the reviewer
meant here. The example images used by our deep learning model are already
reported in Figure 20.  

13. Conditions such as ambient light conditions and size changes in the image adversely affect deep learning results. Did you use any data augmentation methods to avoid these situations?
>No, data augmentation methods have not been used, since an onsite re-
training is likely to be necessary. Therefore, this opens the possibility to have
images representing the real environment and the relative ambient light, which
indeed has an important role, as mentioned in Section 3.2. Nevertheless, the
retraining phase is not expected to take higher effort with respect to the proof-
of-concept, thanks to the good dimensionality ratio between the data set size
and the model size, as well as the generalisation capabilities shown by out neural networks model.  

14. In the article, in addition to error prevention, it was expected to reduce non-value added operations further with improvement suggestions in processes. It is seen that this gap is low. Criticizing this, a comparative evaluation and analysis of the results obtained should be done collectively in a table.
>We honestly did not properly understand this comment. We would be happy
to address it, in case the reviewer could clarify this point. Hopefully all the
other corrections and improvements, following also the suggestions of the other
reviewers, have already contributed to address this issue.  

15. Spelling errors and other errors should be corrected.
>A thorough proofreading of the paper has been performed.

16. Conclusions should be rewritten: references given in the conclusions should be given in the previous sections. In conclusions, mainly the obtained results in the study should be presented, and the limitations of the study and future studies should be explained.
>Conclusions have been re-written following this suggestion.